# Prevalence of Human Papilloma Virus Infection in Bladder Cancer: A Systematic Review

**DOI:** 10.3390/diagnostics12071759

**Published:** 2022-07-20

**Authors:** Narcisa Muresu, Biagio Di Lorenzo, Laura Saderi, Illari Sechi, Arcadia Del Rio, Andrea Piana, Giovanni Sotgiu

**Affiliations:** 1Department of Medicine, Surgery and Pharmacy, University of Sassari, 07100 Sassari, Italy; narcisamuresu@outlook.com (N.M.); illasechi@uniss.it (I.S.); 2Clinical Epidemiology and Medical Statistics Unit, Department of Medical, Surgical and Experimental Medicine, University of Sassari, 07100 Sassari, Italy; dilorbiagio@gmail.com (B.D.L.); lsaderi@uniss.it (L.S.); gsotgiu@uniss.it (G.S.); 3Department of Biomedical Science, University of Sassari, 07100 Sassari, Italy; delrio.arcadia2@gmail.com

**Keywords:** human papillomavirus, HPV, bladder cancer, HPV detection, urothelial carcinoma, transitional cell carcinoma

## Abstract

The etiology of bladder cancer is known to be associated with behavioral and environmental factors. Moreover, several studies suggested a potential role of HPV infection in the pathogenesis with controversial results. A systematic review was conducted to assess the role of HPV. A total of 46 articles that reported the prevalence of HPV infection in squamous (SCC), urothelial (UC), and transitional cell carcinomas (TCC) were selected. A pooled prevalence of 19% was found, with a significant difference in SCC that was mainly driven by HPV-16. Moreover, infection prevalence in case-control studies showed a higher risk of bladder cancer in HPV-positive cases (OR: 7.84; *p*-value < 0.00001). The results may suggest an etiologic role of HPV in bladder cancer. HPV vaccine administration in both sexes could be key to prevent the infection caused by high-risk genotypes.

## 1. Introduction

Bladder cancer (BCa) is the 10th most prevalent cancer globally, with >572,000 and >212,000 incident cases and deaths [1]. The highest rates of BCa are registered in Southern and Western Europe and North America, with a higher incidence in men [2]. Several risk factors were found; most of them are related to personal behavior (i.e., diet and smoking), socioeconomic status (i.e., the accessibility to health services and delay in diagnosis), and environmental and occupational exposure to chemical substances or infectious diseases [3]. Moreover, epidemiological studies revealed a higher risk in men and a worse prognosis in black males [4]. BCa histological subtypes can vary: urothelial BCa, previously classified as transitional cell carcinoma (TCC) and the predominant histological type, accounts for ~90% and is mainly related to chemical exposure, whereas squamous cell carcinoma (SCC, 5%) is associated with chronic inflammation and persistent infections (Schistosoma spp. in Africa) [5]. Adenocarcinoma (2%), sarcoma, and small cell carcinoma are less incident forms [6].

While the role played by human papillomavirus (HPV) in the development of cervical, anogenital, and oropharyngeal cancers was proven [7,8,9], the causative relationship between HPV and BCa still remains controversial, with a high variable prevalence attributed to the study design, the enrolled population, and the HPV detection methods [10]. Understanding the role of HPV in BCa could have relevant diagnostic, therapeutic, and preventive implications. The current systematic review is aimed to assess the prevalence of HPV infection in BCa, focusing on patients’ clinical and epidemiological characteristics.

## 2. Materials and Methods

### 2.1. Search Strategy

A systematic literature review aimed at retrieving papers focused on the prevalence of HPV infection in BCa was carried out from its inception to 31st December 2021. The literature search was performed using PubMed and Scopus, selecting the key words “Human Papillomavirus”, “HPV”, or “Papillomavirus” and “bladder cancer” or “bladder carcinoma”, combined in different strings. No restrictions related to age of patients, setting, or time of the study were chosen.

Lists of references of all selected articles were screened to find other eligible studies not included in the above-mentioned databases.

### 2.2. Study Selection and Inclusion Criteria

Case-control, cross-sectional, and cohort studies reporting HPV infection prevalence were selected.

The following inclusion criteria were considered:
(1)Studies dealing with patients with primary BCa;(2)Studies describing molecular and non-molecular HPV detection methods on fresh or FFPE (Formalin Fixed Paraffin Embedded) bladder biopsies;(3)Studies focused on the following medical conditions: SCC, urothelial carcinoma (UC), and transitional cell carcinoma (TCC);

Articles were excluded for the following reasons:(1)Review articles, abstracts, letters, commentaries, correspondences, case-reports, and case-series enrolling <10 subjects;(2)Use of languages other than English;(3)Secondary malignancies located in the bladder.

Article selection and data extraction were performed by two Authors and double-checked (M.N. and D.B.), while discrepancies of opinions or disagreement were resolved by a third investigator (S.G.).

### 2.3. Data Extraction

Qualitative and quantitative variables were collected in an ad hoc electronic form.

The following variables were collected: first author’s last name; title of the article; year and country/countries of the study; period of the study; epidemiological study design; sample size; sex and age; type of samples; HPV detection methods; histological subtypes; tumor grading; HPV prevalence.

No ethical approval was needed given the anonymized and aggregated nature of the data.

### 2.4. Study Quality Assessment

Inter-rater agreement was ~100% for the phases of study selection and data extraction, and the few inconsistencies were resolved by consensus and with the support of a third investigator (G.S.).

Guidelines of the Preferred Reporting Items for Systematic Reviews and Meta-Analysis (PRISMA) were followed to guide the process of the systematic review [11].

The Newcastle–Ottawa Scale [12] was used to assess the quality of the included case-control studies by evaluating selection, comparability, and exposure criteria, through four, two, and three items, respectively. The Joanna Briggs Institute Critical Appraisal tools (JBI), applied for analytical cross-sectional studies where the control group is missing, consists of eight items aimed at evaluating the risk of bias: high, moderate, or low risk of bias was assigned when positive answers were ≤49%, between 50% and 75%, or >75%, respectively [13] (Appendix A).

### 2.5. Statistical Analysis

Qualitative and quantitative variables were summarized with absolute and relative (percentage) frequencies and means/medians [standard deviation (SD), interquartile range (IQR)] respectively.

Forest plots were used to show pooled risk differences of the selected outcomes and interval (95% confidence interval, CI) estimates, as well as the weight of the sample size of the recruited studies. The I2 indicator (low, medium, and high heterogeneity expressed as <25%, ≥25%–<50%, ≥50%, respectively) showed the association between true variability and overall variation.

Fixed and random-effects models were chosen depending on the estimated between-study heterogeneity. A two-tailed *p*-value less than 0.05 was deemed statistically significant. The statistical software Stata version 17 (StataCorp, College Station, TX, USA) and StatsDirect version 3.1.12 (StatsDirect Ltd., Willar, UK).

## 3. Results

### 3.1. Study Selection

A total of 637 articles were identified through electronic database searches; 162 (26.2%) were excluded for being duplicates, and then, a total of 475 studies were screened by titles and abstracts. Fifty-six (11.8%) full texts were evaluated, and ten (17.9%) were excluded for the following reasons: tissue samples did not include bladder tissues (n = 5), full text was not available (n = 2), case-report (n = 1), review (n = 1), and dataset described in another study (n = 1). A total of 46 (46/56; 82.1%) manuscripts were included in the review (Table 1; Figure 1).

### 3.2. Quality Assessment

Twelve (34.3%) cross-sectional studies were deemed at moderate risk of bias, whereas 23 (65.7%) were classified as low-risk (Table 2). Nine (81.8%) cross-sectional studies were deemed to be medium-quality, whereas two (18.2%) were high-quality (Table 3).

### 3.3. Study Characteristics

Studies were published during the period 1995 [31,35,36,38]–2021 [44,57]. Patients were enrolled between 1985 [17] and 2019 [57]. The epidemiological study types were observational retrospective (26, 56.5%) [15,16,17,18,23,24,25,26,27,29,30,31,34,35,36,38,39,40,44,46,52,55,56,57,58,59], case-control (11, 23.9%) [14,19,20,32,37,42,47,49,50,53,54], prospective (8, 17.4%) [21,22,33,41,43,45,48,51], and comparative (1, 2.2%) [28]. Most of them were single-center (44/46, 95.7%) [14,15,16,19,20,21,22,23,24,25,26,27,28,29,30,31,32,33,34,35,36,37,38,39,40,41,42,43,44,45,46,47,48,49,50,51,52,53,54,55,56,57,58,59], and only two (4.3%) were multi-center [17,18]. Single-center studies were performed in Europe (18, 40.9%) [27,28,29,30,35,39,40,41,45,46,47,48,53,54,55,56,58,59], Asia (14, 31.8%) [14,15,20,23,31,34,36,37,42,43,44,50,51,57], America (6, 13.6%) [16,24,25,32,38,52], and Africa (6, 13.6%) [19,21,22,26,33,49].

### 3.4. Characteristics of the Study Samples

The sample size ranged from 10 [31] to 689 [39] patients, for a total of 3975 subjects. Information on gender was reported by 36 (73.5%) studies [14,15,16,17,19,20,21,22,24,25,26,27,28,29,30,31,32,33,34,37,40,41,42,43,44,45,46,47,48,50,51,53,55,56,57,58], including 555 and 1969 females and males, respectively. The mean/median age ranged from 47 [26] to 74.8 [44] years (Appendix A).

The majority of the samples were FFPE (2706/3518; 76.9%) [14,15,16,17,18,20,21,22,23,24,25,26,30,31,32,33,34,35,36,37,38,40,41,42,43,44,45,46,48,49,50,51,53,55,56,58,59], and only 812 (23.1%) were fresh tissue specimens [19,28,29,42,47,48,50,54,57,59]. Information on the type of specimen was not available for three (3/46; 6.5%) studies [27,39,52].

Histological classification was available for 42/46 (91.3%): TCC were the most prevalent (1445/2792; 51.8%) type, followed by UC (1098/2792; 39.3%) and SCC (249/2792; 8.9%) (Appendix A).

Sixteen (34.8%) studies [14,21,27,30,35,38,40,41,45,48,50,51,52,54,55], for a total of 1428 samples, reported the grading, following the recommendations of the American Joint Committee on Cancer [AJCC Cancer Staging Manual. 7th ed. New York, NY: Springer; 2010.]. A total of 580 (40.6%), 513 (35.9%), and 335 (23.5%) tumors were classified as moderate (G2), poor (G3), and well (G1) differentiated, respectively. 1049 specimens [15,20,22,33,34,36,37,42,43,45,47,49,51,57,58] were classified according to the guidelines of the European Association of Urology [60], with 541 (51.6%) low- and 508 (48.4%) high- grade lesions (Appendix A).

The most frequent HPV detection method was molecular (38/46; 82.6%) [15,19,20,21,22,23,24,26,28,29,30,31,32,33,34,35,36,37,38,39,40,41,42,43,44,45,46,47,48,50,51,52,53,54,55,57,58,59], whereas a non-molecular technique (i.e., immunohistochemistry–IHC and/or in situ hybridization–ISH) was employed in 34 (73.9%) studies [14,16,17,18,23,24,25,27,31,32,33,35,37,41,43,44,46,49,50,51,55,56,59] (Appendix A)

### 3.5. Outcomes

Pooled HPV prevalence was 19% (95% CI: 13%-26%; I2: 96.4%) (Figure 2) ranging from 0% [17,21,26,41,47,52,55,57,58] to 83% [48]. 619/3682 (16.8%) BCa samples were positive (Table 4).

No risk differences were found between females and males [pooled risk difference (95% CI): 0.0046 (−0.0545; 0.0636); *p*-value: 0.87999; I2: 13.5%] (Appendix A). No statistically significant risk differences were found when prevalences related to patients in stage ≤T1 and ≥T2 stage were compared [pooled risk difference (95% CI) = −0.0659 (−0.173; 0.0411); *p*-value: 0.22746; I2: 45.2%], as well as in patients with G1 and G2/G3 tumors, [pooled risk difference (95% CI): −0.0451 (−0.1447; 0.0546); *p*-value: 0.37542; I2: 75.9%] (Appendix A).

Pooled prevalence stratified by histological subtypes was 36.5% (95% CI: 15.9–60.1%; I2: 88.2%), 32.5% (95% CI: 23.8–41.8%; I2: 90.7%), and 18.5% (95% CI: 3.8–40.9%; I2: 97.3%) for SCC, TCC, and UC, respectively (Figure 3), with a statistically significant difference between the prevalence in SCC vs. TCC (*p*-value: 0.0002) and vs. UC (*p*-value < 0.0001).

Also, 69/479 (6.9%) were multiple infections [15,19,20,27,32,35,36,44,50,52,55,57].

The most prevalent genotype was HPV-16 (216/479; 45.1%), followed by HPV-18 (153/479; 31.9%), HPV-6 (25/479; 5.2%), and HPV-11 (17/479; 3.5%). Moreover, 68 (14.2%) infections were caused by other HR-HPV genotypes (Appendix A).

### 3.6. Case-Control Studies

The studies with a case-control design numbered 11/46 (23.9%) in the selection, with a total of 611 cases vs. 227 controls [14,19,20,32,37,42,47,49,50,53,54]. The overall prevalence was 27.3% (167/611) and 4.4% (10/227) in cases and controls, respectively. The pooled odds ratio (OR) was 7.8406 (95% CI, 4.3425–14.1567; *p*-value < 0.00001; I2: 34.7%) for the association between HPV infection and occurrence of BCa (Table 5; Figure 3).

## 4. Discussion

This systematic review was performed to evaluate the prevalence of HPV infection in BCa, keeping into consideration confounding demographic, histological, and diagnostic variables.

An overall HPV prevalence of 19% was found in 46 studies, in line with previous meta-analyses which reported a prevalence of 16.88% [10] and 14.3% [61]. Despite the growing number of reported studies, the role of HPV in cancers other than genital, anal, head, and neck cancers is still debated, due to the heterogeneity in the study design, population enrolled, and HPV detection methods [6]. In fact, several systematic reviews, which evaluated the prevalence of infection of DNA-based vs. non-DNA-based methods confirmed the higher specificity and sensitivity of molecular-based methods [62,63]. Moreover, the use of genotype primers designed for shorter DNA sequences reduced the risk of “false negative” results in comparison with broad-spectrum primers (i.e., GP5+/6+), especially for FFPE specimens often undergoing DNA damage [10]. Therefore, the implementation of a standardized procedure for HPV detection could better clarify the impact of HPV in BCa pathogenesis.

A statistically significant difference in the prevalence of infection was found in different histological subtypes, with higher estimates in SCC than those in TCC and UC. The high affinity of HPV to differentiating squamous epithelium, previously demonstrated in cervical, head and neck, and anogenital carcinomas [64], as well as the ability of the virus to evade and inactivate the immune response, could explain the mechanism of carcinogenesis in the bladder epithelium. Since SCC has poor prognosis and is associated with worse outcomes in different sites [65], the confirmation of the role of HPV in SCC could improve the epidemiological burden and the prognosis of potential cases.

The assessment of the role of HPV as causative agent in different histological subtypes could have been affected by the adoption of the new WHO BCa classification [66]. This recent recommendation changed the traditional nomenclature of “transitional cell carcinoma” in “urothelial carcinoma”, causing potential misinterpretations of the results. On this basis, the identification of standard accurate procedures for the detection and diagnosis of HPV infection could be helpful to estimate the burden of cancers associated with HPV. However, the available molecular techniques show a high diagnostic accuracy.

Although several studies suggested a relationship between advanced stage (i.e., ≥T2) and HPV prevalence [27,43,44], the poor information collected in the selected studies did not prove this association.

Similarly to other HPV-related cancers, HPV-16 was the most prevalent genotype in BCa, supporting previous findings on the increased risk of BCa in cases of infection caused by high-risk HPV genotypes, mainly by HPV-16 [10]. The high detection rate of HPV-16, together with HPV-18 and the low-risk HPV-6, strongly supports the administration of HPV vaccines to prevent HPV-related cancers in both sexes [67].

Finally, our study showed a significant association between HPV and BCa (OR: 7.84), confirming the findings of recent meta-analyses on the risk of BCa [10,62,65]. However, these results are in contrast with a previous study published by Khatami et al. [61], who described a non-significant association between infection and cancer and highlighted the scientific need of larger case-control studies. Although the majority of case-control studies were classified as “medium-quality”, controls showed a prevalence of infection <10%, regardless of HPV detection methods, suggesting the etiological role of HPV in BCa.

Previous systematic reviews investigated the role of HPV infection in BCa and described a moderate association. Our study selection and analysis showed an important role of HPV infection in SCC BCa. However, some limitations should be acknowledged. More stratified analyses related to demographic (i.e., geographical area, behavioral factors, occupational exposures) and clinical (i.e., stage, HPV detection methods, prognosis of patients) confounders should be performed. The cross-sectional assessment does not help prove the temporal relationship between an exposure and the development of cancer.

## 5. Conclusions

The present systematic review highlights a potential role of HPV in the development of bladder cancer, indirectly supporting the adoption of primary preventive strategies in both sexes, as recommended by international authorities. Further epidemiological studies are needed to confirm those findings and assess the role of diagnostic and preventive strategies for HPV-related bladder cancer.

## Figures and Tables

**Figure 1 diagnostics-12-01759-f001:**
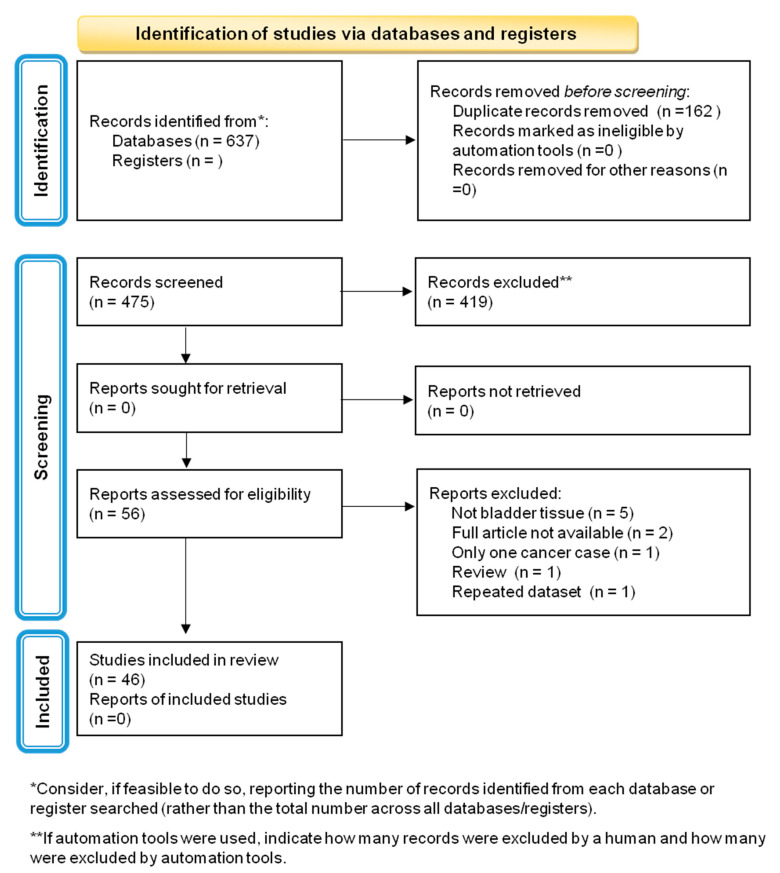
PRISMA 2020 flow diagram for new systematic reviews, which included searches of databases and registers only.

**Figure 2 diagnostics-12-01759-f002:**
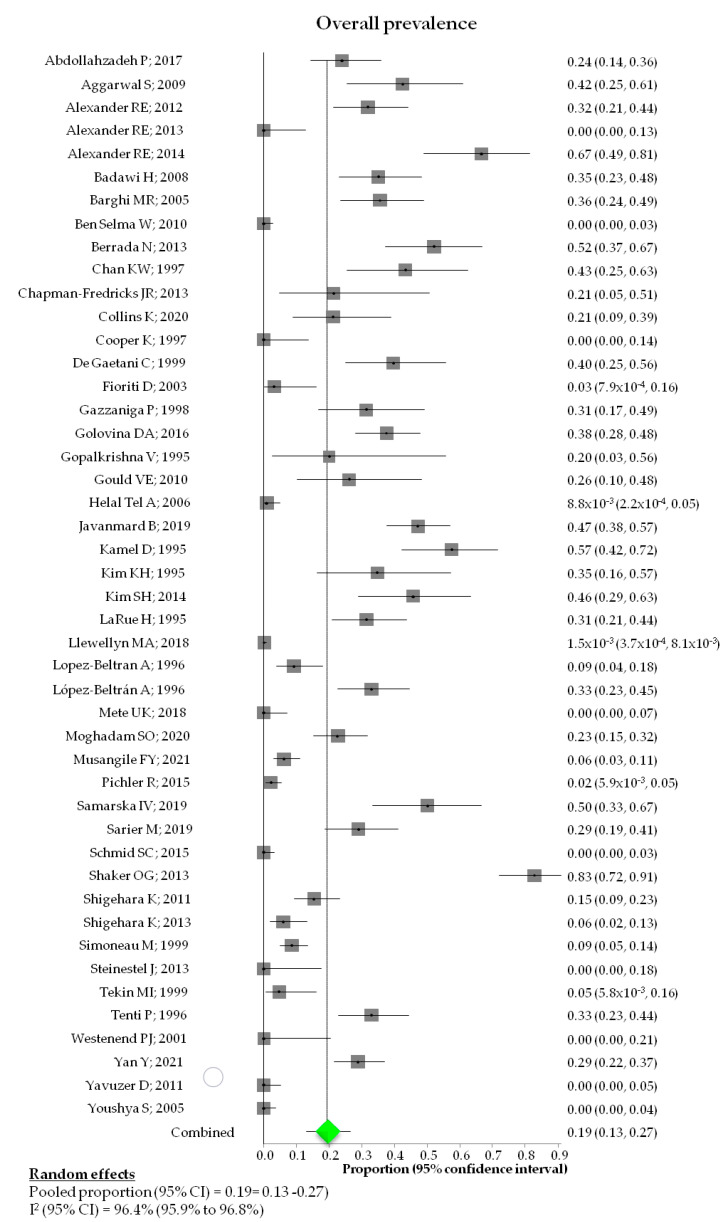
Forest plot of HPV pooled prevalence in bladder cancer. Adapted from [14,15,16,17,18,19,20,21,22,23,24,25,26,27,28,29,30,31,32,33,34,35,36,37,38,39,40,41,42,43,44,45,46,47,48,49,50,51,52,53,54,55,56,57,58,59].

**Figure 3 diagnostics-12-01759-f003:**
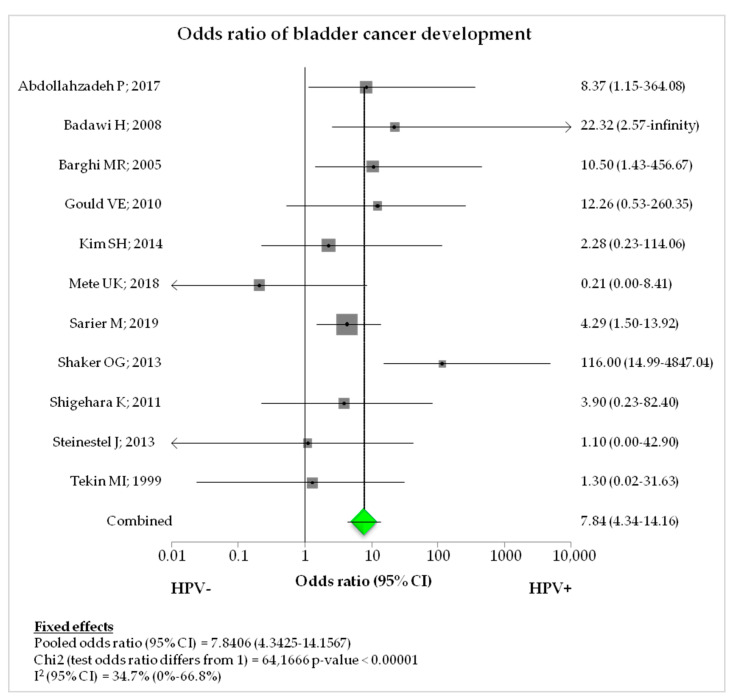
Forest plot of overall odds ratio in case-control studies. Adapted from [14,19,20,32,37,42,47,49,50,53,54].

**Table 1 diagnostics-12-01759-t001:** Characteristics of the included studies (n = 46).

Ref.	First Author	Year	Title	Type of Study	Multicentre/Single Centre	Country/Ies	Study Period
[14]	Abdollahzadeh P, et al.	2017	Association Between Human Papillomavirus and Transitional Cell Carcinoma of the Bladder	Case/control study	Single	Iran	2008–2011
[15]	Aggarwal S, et al.	2009	Koilocytosis: correlations with high-risk HPV and its comparison on tissue sections and cytology, urothelial carcinoma	Retrospective observational study	Single	India	-
[16]	Alexander RE, et al.	2012	p16 expression is not associated with human papillomavirus in urinary bladder squamous cell carcinoma	Retrospective observational study	Single	USA	1992–2011
[17]	Alexander RE, et al.	2013	Human papillomavirus is not an etiologic agent of urothelial inverted papillomas	Retrospective observational study	Multi	USA Spain Italy France	1985–2005
[18]	Alexander RE, et al.	2014	The expression patterns of p53 and p16 and an analysis of a possible role of HPV in primary adenocarcinoma of the urinary bladder	Retrospective observational study	Multi	USA Spain Italy France	-
[19]	Badawi H, et al.	2008	Role of human papillomavirus types 16, 18, and 52 in recurrent cystitis and urinary bladder cancer among Egyptian patients	Case/control study	Single	Egypt	2001–2006
[20]	Barghi MR, et al.	2005	Correlation between human papillomavirus infection and bladder transitional cell carcinoma	Case/control study	Single	Iran	1999–2002
[21]	Ben Selma W, et al.	2010	Investigation of human papillomavirus in bladder cancer in a series of Tunisian patients	Observational study	Single	Tunisia	2003–2004
[22]	Berrada N, et al.	2013	Human papillomavirus detection in Moroccan patients with bladder cancer	Prospective study	Single	Morocco	-
[23]	Chan KW, et al.	1997	Prevalence of six types of human papillomavirus in inverted papilloma and papillary transitional cell carcinoma of the bladder: an evaluation by polymerase chain reaction	Retrospective observational study	Single	China	1987–1994
[24]	Chapman-Fredricks JR, et al.	2013	High-risk human papillomavirus DNA detected in primary squamous cell carcinoma of urinary bladder	Retrospective observational study	Single	USA	-
[25]	Collins K, et al.	2020	Prevalence of high-risk human papillomavirus in primary squamous cell carcinoma of urinary bladder	Retrospective observational study	Single	Texas	2009–2019
[26]	Cooper K, et al.	1997	Human papillomavirus and schistosomiasis associated bladder cancer	Retrospective observational study	Single	South Africa	-
[27]	De Gaetani C, et al.	1999	Detection of human papillomavirus DNA in urinary bladder carcinoma by in situ hybridisation	Retrospective observational study	Single	Italy	1995–1997
[28]	Fioriti D, et al.	2003	Urothelial bladder carcinoma and viral infections: different association with human polyomaviruses and papillomaviruses	Comparative study	Single	Italy	-
[29]	Gazzaniga P, et al.	1998	Prevalence of papillomavirus, Epstein-Barr virus, cytomegalovirus, and herpes simplex virus type 2 in urinary bladder cancer	Retrospective observational study	Single	Italy	-
[30]	Golovina DA, et al.	2016	Loss of Cell Differentiation in HPV-Associated Bladder Cancer	Retrospective observational study	Single	Russia	-
[31]	Gopalkrishna V, et al.	1995	Detection of human papillomavirus DNA sequences in cancer of the urinary bladder by in situ hybridisation and polymerase chain reaction	Retrospective observational study	Single	India	-
[32]	Gould VE, et al.	2010	Human papillomavirus and p16 expression in inverted papillomas of the urinary bladder	Case/control study	Single	USA	-
[33]	Helal Tel A, et al.	2006	Human papilloma virus and p53 expression in bladder cancer in Egypt: relationship to schistosomiasis and clinicopathologic factors	Observational study	Single	Egypt	-
[34]	Javanmard B, et al.	2019	Human Papilloma Virus DNA in Tumor Tissue and Urine in Different Stage of Bladder Cancer	Retrospective observational study	Single	Iran	2014–2016
[35]	Kamel D, et al.	1995	Human papillomavirus DNA and abnormal p53 expression in carcinoma of the urinary bladder	Retrospective observational study	Single	Finland	1987–1992
[36]	Kim KH, et al.	1995	Analysis of p53 tumor suppressor gene mutations and human papillomavirus infection in human bladder cancers	Retrospective observational study	Single	Korea	-
[37]	Kim SH, et al.	2014	Detection of human papillomavirus infection and p16 immunohistochemistry expression in bladder cancer with squamous differentiation	Case/control study	Single	Korea	2001–2011
[38]	LaRue H, et al.	1995	Human papillomavirus in transitional cell carcinoma of the urinary bladder	Retrospective observational study	Single	Canada	-
[39]	Llewellyn MA, et al.	2018	Defining the frequency of human papillomavirus and polyomavirus infection in urothelial bladder tumours	Retrospective observational study	Single	UK	2005–2011
[40]	Lopez-Beltran A, et al.	1996a	Human papillomavirus DNA as a factor determining the survival of bladder cancer patients	Retrospective observational study	Single	Spain	-
[41]	López-Beltrán A, et al.	1996b	Human papillomavirus infection and transitional cell carcinoma of the bladder: Immunohistochemistry and in situ hybridization	Observational study	Single	Spain	-
[42]	Mete UK, et al.	2018	Human Papillomavirus in Urothelial Carcinoma of Bladder: An Indian study	Case/control study	Single	India	-
[43]	Moghadam SO, et al.	2020	Association of human papilloma virus (HPV) infection with oncological outcomes in urothelial bladder cancer	Prospective study	Single	Iran	-
[44]	Musangile FY, et al.	2021	Detection of HPV infection in urothelial carcinoma using RNAscope: Clinicopathological characterization	Retrospective observational study	Single	Japan	2013–2019
[45]	Pichler R, et al.	2015	Low prevalence of HPV detection and genotyping in non-muscle invasive bladder cancer using single-step PCR followed by reverse line blot	Prospective study	Single	Austria	-
[46]	Samarska IV, et al.	2019	Condyloma Acuminatum of Urinary Bladder: Relation to Squamous Cell Carcinoma	Observational study	Single	*	-
[47]	Sarier M, et al.	2019	Is There any Association between Urothelial Carcinoma of the Bladder and Human Papillomavirus? A Case-Control Study	Case/control study	Single	Turkey	Jan–Dec 2018
[48]	Schmid SC, et al.	2015	Human papilloma virus is not detectable in samples of urothelial bladder cancer in a central European population: a prospective translational study	Prospective study	Single	Germany	-
[49]	Shaker OG, et al.	2013	Is there a correlation between HPV and urinary bladder carcinoma?	Case/control study	Single	Egypt	-
[50]	Shigehara K, et al.	2011	Etiologic role of human papillomavirus infection in bladder carcinoma	Case/control study	Single	Japan	1997–2009
[51]	Shigehara K, et al.	2013	Etiological correlation of human papillomavirus infection in the development of female bladder tumor	Prospective study	Single	Japan	1996–2010
[52]	Simoneau M, et al.	1999	Low frequency of human papillomavirus infection in initial papillary bladder tumors	Retrospective observational study	Single	Canada	1990–1992
[53]	Steinestel J, et al.	2013	Overexpression of p16INK4a in Urothelial Carcinoma In Situ Is a Marker for MAPK-Mediated Epithelial-Mesenchymal Transition but Is Not Related to Human Papillomavirus Infection	Case/control study	Single	Germany	2001–2011
[54]	Tekin MI, et al.	1999	Human papillomavirus associated with bladder carcinoma? Analysis by polymerase chain reaction	Case/control study	Single	Turkey	-
[55]	Tenti P, et al.	1996	p53 overexpression and human papillomavirus infection in transitional cell carcinoma of the urinary bladder: correlation with histological parameters	Retrospective observational study	Single	Italy	-
[56]	Westenend PJ, et al.	2001	Human papillomaviruses 6/11, 16/18 and 31/33/51 are not associated with squamous cell carcinoma of the urinary bladder	Retrospective observational study	Single	Netherlands	-
[57]	Yan Y, et al.	2021	Human Papillomavirus Prevalence and Integration Status in Tissue Samples of Bladder Cancer in the Chinese Population	Retrospective observational study	Single	China	2015–2019
[58]	Yavuzer D, et al.	2011	Role of human papillomavirus in the development of urothelial carcinoma	Retrospective observational study	Single	Turkey	-
[59]	Youshya S, et al.	2005	Does human papillomavirus play a role in the development of bladder transitional cell carcinoma? A comparison of PCR and immunohistochemical analysis	Retrospective observational study	Single	England	-

* not specified.

**Table 2 diagnostics-12-01759-t002:** JBI risk of bias assessment table. Eight items per study were evaluated and the risk of bias was calculated on the number of positive answers. y = yes, n = no, u = unclear. Moderate = where positive answers were between 50% and 75%; low = where positive answers were above 75%.

Ref.	First Author	Were the Criteria for Inclusion in the Sample Clearly Defined?	Were the Study Subjects and the Setting Described in Detail?	Was the Exposure Measured in a Valid and Reliable Way?	Were Objective, Standard Criteria Used for Measurement of the Condition?	Were Confounding Factors Identified?	Were Strategies to Deal with Confounding Factors Stated?	Were the Outcomes Measured in a Valid and Reliable Way?	Was Appropriate Statistical Analysis Used?	% YES	Risk
[15]	Aggarwal S; 2009	y	y	y	y	y	n	y	u	75	Low
[16]	Alexander RE; 2012	y	y	y	y	y	n	y	n	75	Low
[17]	Alexander RE; 2013	y	y	y	y	y	n	y	n	75	Low
[18]	Alexander RE; 2014	y	y	y	y	y	n	y	n	75	Low
[21]	Ben Selma W; 2010	y	y	y	y	y	n	y	n	75	Low
[22]	Berrada N; 2013	y	y	y	y	y	y	y	n	88	Low
[23]	Chan KW; 1997	y	y	y	y	n	n	y	y	75	Low
[24]	Chapman-Fredricks JR; 2013	y	y	y	y	y	y	y	n	88	Low
[25]	Collins K; 2020	y	y	y	y	y	y	y	y	100	Low
[26]	Cooper K; 1997	y	n	y	y	y	n	y	n	63	Moderate
[27]	De Gaetani C; 1999	y	n	y	y	y	n	y	y	75	Low
[28]	Fioriti D; 2003	y	n	y	n	y	n	y	n	50	Moderate
[29]	Gazzaniga P; 1998	y	n	y	y	y	y	y	n	75	Low
[30]	Golovina DA; 2016	y	y	y	y	y	y	y	y	100	Low
[31]	Gopalkrishna V; 1995	y	n	y	n	y	y	y	n	63	Moderate
[33]	Helal Tel A; 2006	y	y	y	n	y	y	y	y	88	Low
[34]	Javanmard B; 2019	y	n	y	n	y	n	y	y	63	Moderate
[35]	Kamel D; 1995	y	n	y	y	n	n	y	n	50	Moderate
[36]	Kim KH; 1995	y	n	y	y	n	n	y	n	50	Moderate
[38]	LaRue H; 1995	y	y	y	y	n	n	n	n	50	Moderate
[39]	Llewellyn MA; 2018	y	y	y	y	n	n	y	n	63	Moderate
[40]	Lopez-Beltran A; 1996a	y	y	y	y	y	y	y	y	100	Low
[41]	López-Beltrán A; 1996b	y	y	y	y	y	y	y	n	88	Low
[43]	Moghadam SO; 2020	y	y	y	y	y	y	y	y	100	Low
[44]	Musangile FY; 2021	y	y	y	y	y	n	y	y	88	Low
[45]	Pichler R; 2015	y	y	y	y	y	y	y	y	100	Low
[46]	Samarska IV; 2019	y	y	y	n	y	y	y	n	75	Low
[48]	Schmid SC; 2015	y	y	y	y	y	n	y	n	75	Low
[51]	Shigehara K; 2013	y	n	y	y	y	y	y	y	88	Low
[52]	Simoneau M; 1999	y	n	y	y	n	n	y	n	50	Moderate
[55]	Tenti P; 1996	y	y	y	y	y	n	y	n	75	Low
[56]	Westenend PJ; 2001	y	n	y	y	y	n	y	n	63	Moderate
[57]	Yan Y; 2021	y	n	y	y	y	y	y	y	88	Low
[58]	Yavuzer D; 2011	y	y	y	y	n	n	y	n	63	Moderate
[59]	Youshya S; 2005	y	n	y	y	y	n	y	n	63	Moderate

**Table 3 diagnostics-12-01759-t003:** NOS quality assessment table. Each study was awarded one star per item within the selection and exposure categories. A maximum of two stars could be awarded for comparability. The score is the sum of the awarded stars and ranges from zero to nine.

Ref.	First Author	Selection				Comparability	Exposure			Score
		Is the Case Definition Adequate?	Representativeness of the Cases	Selection of Controls	Definition of Controls	Based on the Design or Analysis	Ascertainment of Exposure	Same Method of Ascertainment for Cases and Controls	Non-Response Rate	
[14]	Abdollahzadeh P; 2017	*	*	*	*	*		*	*	5/9
[19]	Badawi H; 2008	*	*	*	*	*	*	*	*	7/9
[20]	Barghi MR; 2005	*	*	*	*	*	*	*	*	6/9
[32]	Gould VE; 2010	*	*	*	*	*	*	*	*	6/9
[37]	Kim SH; 2014	*	*	*	*	*	*	*	*	6/9
[42]	Mete UK; 2018	*	*	*	*	*		*	*	5/9
[47]	Sarier M; 2019	*	*	*	*	*		*	*	5/9
[49]	Shaker OG; 2013	*	*	*	*	*		*	*	5/9
[50]	Shigehara K; 2011	*	*	*	*	*		*	*	5/9
[53]	Steinestel J; 2013	*	*	*	*	*		*	*	5/9
[54]	Tekin MI; 1999	*		*	*		*	*		5/9

“*^”^: Each study was awarded one star per item. The score is the sum of the awarded stars, ranged from 0 to 9 (high-quality, >7 stars; medium-quality, 4–6 stars; poor-quality, <4 stars).

**Table 4 diagnostics-12-01759-t004:** Overall HPV prevalence among the selected studies. If different prevalence values were outlined in the studies, the value from the standard detection technique (or the highest in case of comparison between standard methods) was reported in the table below.

Ref.	First Author	Overall HPV Prevalence	Ref.	First Author	Overall HPV Prevalence
[14]	*Abdollahzadeh P; 2017*	16/67 (23.9)	[37]	*Kim SH; 2014*	16/35 (45.7)
[15]	*Aggarwal S; 2009*	14/33 (42.4)	[38]	*LaRue H; 1995*	22/70 (31.4)
[16]	*Alexander RE; 2012*	22/69 (31.9)	[39]	*Llewellyn MA; 2018*	1/689 (0.1)
[17]	*Alexander RE; 2013*	0/27 (0)	[40]	*Lopez-Beltran A; 1996*	7/76 (9.2)
[18]	*Alexander RE; 2014*	24/36 (67)	[41]	*Lopez-Beltran A; 1996*	25/76 (32.9)
[19]	*Badawi H; 2008*	21/60 (35)	[42]	*Mete UK; 2018*	0/50 (0)
[20]	*Barghi MR; 2005*	21/59 (35.6)	[43]	*Moghadam SO; 2020*	24/106 (22.6)
[21]	*Ben Selma W; 2010*	0/125 (0)	[44]	*Musangile FY; 2021*	10/162 (6.2)
[22]	*Berrada N; 2013*	25/48 (52.1)	[45]	*Pichler R; 2015*	4/186 (2.2)
[23]	*Chan KW; 1997*	13/30 (43.3)	[46]	*Samarska IV; 2019*	19/38 (50)
[24]	*Chapman-Fredricks JR; 2013*	3/14 (21.43)	[47]	*Sarier M; 2019*	20/69 (29)
[25]	*Collins K; 2020*	7/33 (21.2)	[48]	*Schmid SC; 2015*	0/109 (0)
[26]	*Cooper K; 1997*	0/25 (0)	[49]	*Shaker OG; 2013*	58/70 (82.9)
[27]	*De Gaetani C; 1999*	17/43 (32.56)	[50]	*Shigehara K; 2011*	18/117 (15.38)
[28]	*Fioriti D; 2003*	1/32 (3.1)	[51]	*Shigehara K; 2013*	5/84 (5.95)
[29]	*Gazzaniga P; 1998*	11/35 (36.7)	[52]	*Simoneau M; 1999*	16/187 (8.5)
[30]	*Golovina DA; 2016*	38/101 (37.6)	[53]	*Steinestel J; 2013*	0/19 (0)
[31]	*Gopalkrishna V; 1995*	2/10 (20)	[54]	*Tekin MI; 1999*	2/42 (4.8)
[32]	*Gould VE; 2010*	6/23 (26.1)	[55]	*Tenti P; 1996*	26/79 (32.9)
[33]	*Helal Tel A; 2006*	1/114 (0.9)	[56]	*Westenend PJ; 2001*	0/16 (0)
[34]	*Javanmard B; 2019*	52/110 (47.3)	[57]	*Yan Y; 2021*	42/146 (28.8)
[35]	*Kamel D; 1995*	27/47 (57)	[58]	*Yavuzer D; 2011*	0/70 (0)
[36]	*Kim KH; 1995*	8/23 (34.7)	[59]	*Youshya S; 2005*	0/98 (0)

**Table 5 diagnostics-12-01759-t005:** HPV prevalence stratified by the presence (case) or absence (control) of any type of bladder tumor (case-control studies).

Ref.	First Author	HPV Positivity (n/N, %)	
		Cases	Controls
[14]	*Abdollahzadeh P; 2017*	15/67 (22.4)	1/30 (3.3)
[19]	*Badawi H; 2008*	21/60 (35)	0/20 (0)
[20]	*Barghi MR; 2005*	21/59 (35.6)	1/20 (5)
[32]	*Gould VE; 2010*	6/23 (26.1)	0/10 (0)
[37]	*Kim SH; 2014*	6/35 (17.1)	1/12 (8.3)
[42]	*Mete UK; 2018*	0/50 (0)	0/10 (0)
[47]	*Sarier M; 2019*	20/69 (28.9)	6/69 (8.7)
[49]	*Shaker OG; 2013*	58/70 (82.9)	1/25 (4)
[50]	*Shigehara K; 2011*	18/117 (15.4)	0/10 (0)
[53]	*Steinestel J; 2013*	0/19 (0)	0/21 (0)
[54]	*Tekin MI; 1999*	2/42 (4.8)	0/10 (0)

## Data Availability

The data is available in case it is requested for motivated reasons.

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
