# Peer review of "Prevalence of Human Papilloma Virus Infection in Bladder Cancer: A Systematic Review"

_diagnostics, 2022, doi:10.3390/diagnostics12071759_

Round 1
Reviewer 1 Report
It is with great pleasure that I reviewed the article entitled “Prevalence of human papilloma virus infection in bladder cancer: a systemic review” by Muresu et al. In this article, the authors report their systematic review of published literature on HPV and bladder cancer association. A pooled prevalence of 19% was found, with a significant difference in SCC, mainly driven by HPV-16. Moreover, infection prevalence in case-control studies showed a higher risk of bladder cancer in HPV-positive cases (OR:7.84; p-value <0.00001). Being a systemic review, its quality is based on the quality of the available articles. This is a well written, well laid out manuscript. Below is a pint by point critique of the manuscript.
TITLE: No issues
ABSTRACT: No issues
INTRODUCTION: Higher risk of cancer in black men?
MATERIALS AND METHODS:
Limited key words in the search
Dates pulled?
2 consistent reviewers with a 3rd to break ties.
PRIMSA, Newcastle Scale and JBI used.
Was smoking data or exposure data collected and taken into consideration?
RESULTS:
46 studies included in the review with nearly 4,000 subjects.
TCC was the most common histology followed by UC then SCC. But what is the difference between TCC and UC? TCC is the older terminology for UC.
Tumor grade only reported in 34.8% of subjects? This is very low.
How about stage?
What do you mean the most common HPV detection method was molecular? Is there a gold standard to detect HPV? How many followed this gold standard?
83% prevalence of + HPV in bladder tumors?
No difference in male/female.
HPV-16 and HPV-18 were the most common (45% and 32% respectively)
DISCUSSIONS:
SCC may have worse outcomes because it is detecting later. Stage for stage outcomes tend to be the same.
Provide more detail on the pros and cons of your study.
What are the next steps as speculation still remain?
The authors conclude ‘The results support the etiologic role of HPV in bladder cancer.’ I think this needs to be tempered a bit and say ‘The results may suggest an etiologic role of HPV in bladder cancer.’
REFERENCES: No issues
TABLES:
Where’s Table 1?
Table 5-7 and 9
FIGURES: No issues
Reviewer 2 Report
The aim of the article “Prevalence of Human Papilloma Virus infection in bladder cancer: a systematic review” was to determine the role of Human Papilloma Virus in bladder cancer development since potential role of HPV infection in the pathogenesis of bladder cancer is controversial. The article assessed the prevalence of HPV infection in BCa through a systematic revision. The theme is relevant and the results interesting, therefore, the article should be published. However, some points should be clarified. Some questions and suggestions are indicated below:
- Only two databases, PubMed and Scopus were used, discarding other important databases such as Science Direct. Please explain why these two databases were selected.
- Keywords and the combination of search terms for each database (with specific descriptors) should be better specified, since this information allows replication of the searching.
- The paragraph “However, the assessment of the role of HPV as causative agent in different histological subtypes could have been affected by the adoption of the new WHO BCa classification [67] which changed the traditional nomenclature of “transitional cell carcinoma” in “urothelial carcinoma”, causing potential misinterpretations of the results. addressed to the identification of standard procedure for the detection of HPV, could be helpful” is a very important point of the discussion. However, it is too long and confusing, therefore I recommend rewriting it.
- In the “Study selection” section, please write “59” in words, since it is a number after a period. Same comment for “12” in “Quality assessment” section and “16” in “Characteristics of the study samples” section. Please revise these details in all the document.
- “in situ” should be written in italic.
